# Towards Personalized Allele-Specific Antisense Oligonucleotide Therapies for Toxic Gain-of-Function Neurodegenerative Diseases

**DOI:** 10.3390/pharmaceutics14081708

**Published:** 2022-08-16

**Authors:** Jacob Helm, Ludger Schöls, Stefan Hauser

**Affiliations:** 1German Center for Neurodegenerative Diseases (DZNE), 72076 Tübingen, Germany; 2Hertie Institute for Clinical Brain Research and Department of Neurology, University of Tübingen, 72076 Tübingen, Germany; 3Graduate School of Cellular and Molecular Neuroscience, University of Tübingen, 72076 Tübingen, Germany

**Keywords:** antisense oligonucleotide, ASO, neurodegenerative diseases, toxic gain-of-function, allele-specific targeting, SNP

## Abstract

Antisense oligonucleotides (ASOs) are single-stranded nucleic acid strings that can be used to selectively modify protein synthesis by binding complementary (pre-)mRNA sequences. By specific arrangements of DNA and RNA into a chain of nucleic acids and additional modifications of the backbone, sugar, and base, the specificity and functionality of the designed ASOs can be adjusted. Thereby cellular uptake, toxicity, and nuclease resistance, as well as binding affinity and specificity to its target (pre-)mRNA, can be modified. Several neurodegenerative diseases are caused by autosomal dominant toxic gain-of-function mutations, which lead to toxic protein products driving disease progression. ASOs targeting such mutations—or even more comprehensively, associated variants, such as single nucleotide polymorphisms (SNPs)—promise a selective degradation of the mutant (pre-)mRNA while sparing the wild type allele. By this approach, protein expression from the wild type strand is preserved, and side effects from an unselective knockdown of both alleles can be prevented. This makes allele-specific targeting strategies a focus for future personalized therapies. Here, we provide an overview of current strategies to develop personalized, allele-specific ASO therapies for the treatment of neurodegenerative diseases, such Huntington’s disease (HD) and spinocerebellar ataxia type 3 (SCA3/MJD).

## 1. Introduction

Antisense oligonucleotides (ASO) are short, modified single-stranded DNA, RNA, or hybrid DNA-RNA sequences that bind complementary cellular RNAs, such as (pre-)mRNAs, or noncoding RNAs, such as microRNAs, thereby influencing their further processing. In the central dogma of protein synthesis, such an ASO–(pre-)mRNA interaction would prevent the translation of a potentially toxic gene variant. Continuous new modifications of ASOs at their sugar, backbone, or base aim to improve pharmacokinetics and dynamics (reviewed in [1,2,3,4,5]). Thereby, it becomes possible to develop ASOs that target the (pre-)mRNA in an allele-specific manner. This can be achieved by, e.g., selectively targeting single nucleotide polymorphisms (SNP) on the mutant allele, thus providing an approach for a personalized allele-specific treatment [6,7].

Huntington’s disease (HD), amyotrophic lateral sclerosis (ALS), Parkinson’s disease (PD), Alzheimer’s disease (AD) and spinocerebellar ataxia (SCA) are neurodegenerative diseases in which a toxic version of a protein can lead to abnormal protein aggregation [8,9,10]. In such cases, an altered sequence or post-translational modification leads to a conformational or structural change and an increased risk for aggregation and mislocalization [11,12,13,14]. Even though the cause of several monogenetic neurodegenerative diseases driven by toxic gain-of-function mechanisms has known for years or decades [15,16,17,18], there is so far no cure, and current treatments only mitigate specific symptoms, but do not stop or slow down the progression of the disease [19]. One reason why no suitable therapy options have been found so far might be that the toxic gain-of-function often leads to a misprocessing of the protein, which manifests as an aggregation that impairs multiple cellular pathways, leading to neurotoxic effects [11,20,21].

In monogenetic disorders with dominant inheritance, a genetic diagnosis can potentially be made early in life, even before a clinical manifestation of the disease. This offers, in principle, the chance to prevent disease manifestation, if treatment becomes available that “silences” the toxic gain-of-function mutations before the disease-causing effects can no longer be offset.

One possibility to develop a causal therapy could be to degrade the disease protein. This could ideally involve strategies to selectively degrade the toxic protein variant while preserving the wild type form of the protein. ASOs could play a key role in such a therapeutic strategy. In autosomal dominant inherited neurodegenerative diseases with toxic gain-of-function mutations, minor differences between wild type and pathogenic alleles (e.g., SNPs) could be used to selectively degrade the mutant (pre-)mRNA and by this, prevent expression of the toxic protein variant while preserving expression of the healthy copy from the wild type allele [6,7]. Here, we describe the current state and possibilities of personalized, allele-specific research and treatment in neurodegenerative diseases caused by toxic gain-of-function mutations.

## 2. Antisense Oligonucleotides

Antisense oligonucleotides are composed of a strand of DNA or RNA as an initial structure that binds via Watson–Crick hybridization to its complementary RNA strand. The internalization of ASOs is not necessarily linked to a single pathway. Uptake is mostly receptor-dependent with subsequent intracellular trafficking from early endosomes to late endosomes, Golgi apparatus, and lysosomes. However, to fulfill its function in the cytoplasm and nucleus, the ASO must escape the endosome in a process that is likely associated with endosomal transport proteins [22,23]. A similar transport capability for ASOs is not described for the blood–brain barrier (BBB), making it possible to specifically target the brain by intrathecal application [3]. Length and nucleotide arrangement define the mode of action, as well as the target specificity. The arrangement of nucleotides can be a modified RNA strand, a mixmer design, with various alignments of DNA and RNA nucleotides, or a gapmer design, with a central DNA area that is flanked by RNA-based nucleotide wings [24,25,26,27]. Generally, two modes of action can be distinguished (Figure 1A). In the first, modified RNA strands and mixmers act as a steric block. They bind to their target pre-mRNA and thus prevent further processing or modulate splicing [28] (Figure 1B). Binding to complementary pre-mRNA at splice sites can lead to either exon skipping or exon inclusion. Therefore, ASOs can specifically be used to generate a spliced mRNA lacking exons, resulting in frameshifts, mismatches, or early stop codons leading to nonsense-mediated mRNA decay (NMD). In addition, deep intronic splice variants and non-productive transcripts that are pathologically integrated can be targeted by this approach [3,29,30]. The second mode of action of ASOs is via RNase H-dependent degradation. For this, gapmers consisting of flanking RNA wings with a central DNA portion are mainly used. Binding of DNA to its target (pre-)mRNA in the nucleus or cytoplasm recruits RNase H and leads to the degradation of the complementary counterpart [3,30,31,32,33], (Figure 1C). By modifying the backbone, sugar, and/or base of the designed ASO, its characteristics, including stability, longevity, toxicity, target affinity, and specificity, can be modulated (Table 1).

## 3. Backbone Modifications

The most common modification of the backbone is a replacement of a phosphodiester (PO) with a phosphorothioate (PS) bond (Figure 2). This increases nuclease stability and cellular uptake; in addition, the interaction with albumin allows the ASO to be transported within the cerebrospinal fluid (CSF) or blood plasma [61,62,63]. ASO–protein bindings are particularly important with regard to pharmacokinetic properties, since these interactions can delay renal clearance [64]. Since this clearing mechanism is independent from the brain, the main focus on backbone modifications of ASOs for treatments of the central nervous system is on the improved uptake and transport associated with protein binding [61]. The interaction with intracellular proteins is potentially associated with an increase in toxicity, which can be reduced by a site-specific introduction of a neutral PO linkage at position 2 or 3 in the DNA gap [35,36]. However, this modification also results in a reduction of affinity, with a reduction in melting temperature of −0.45 to −1 °C per nucleotide [34]. Another aspect of a sulfur modification is that the negative charge is exclusively on the sulfur atom. This leads to different possible chiralities between the phosphorus and the sulfur atoms [65]. The effects of stereoisomerism of PS on the stability and target cleavage site are currently under debate.

RNase H interacts with the DNA:RNA junction at three consecutive nucleotides. Iwamoto et al. proposed a stereochemical PS backbone of 3′-*S*p*S*p*R*p-5′ in the center of the DNA gapmer as the most suitable site, with the cleavage site located two nucleotides upstream from the *R*p with respect to the DNA [66]. This pattern and its capabilities could not be confirmed by Østergaard and colleagues. However, they showed that a single *R*p junction at position 7 in a *S*p DNA gap, resulted in a single cleavage site, located two nucleotides downstream on the DNA [37]. The potential to predefine cleavage sites could increase the focus for nucleotide-specific modifications to specific positions, thus increasing the impact of modifications at specific sites.

A relatively novel modification of the backbone is the use of mesylphosphoramidate (MsPA) [67,68]. Stetsenko et al. described positive effects on RNase H activity when DNA-ASOs were modified with MsPA compared to PS-modified DNA-ASOs [39]. Ionis Pharmaceuticals could not confirm these positive effects when DNA gapmer ASOs were uniformly modified with MsPA. Slight improvement in RNase H activity could be observed when a maximum of 5 PS bonds in the DNA gap were replaced by MsPA, or Gap 2′-OMe PS ASOs were modified with MsPA at two positions in the 3′ wing, or the 5′ DNA gap [40,69]. Additionally, MsPA positively influences toxicity and half-life of ASOs. An improvement in the stability of nucleases could be obtained when 4 PS bonds of the DNA gap were replaced by MsPA. Even more important might be the effect of MsPA on the interactions with proteins. These interactions could already be substantially reduced by 2 MsPA instead of PS in the DNA gap and could thus become particularly important if protein-associated toxicity needs to be reduced [39,40]. A recent publication indicates the interaction between the design of ASOs and the ionotropic glutamate α-amino-3-hydroxy-5-methyl-4-isoxazolepropionic acid receptor (AMPAR). It has been shown that an increasing number of G bases in an ASO gapmer increases the risk of AMPAR-associated neurotoxicity [70]. Therefore, it will be interesting to see if MsPA modifications around the Gs have an impact on the interaction with AMPAR, thereby modifying potential toxicity. In vivo studies showed a superior effect of MsPA over PS in a fully modified ASO, when delivered in folate-containing liposomes. MsPA showed an improved tumor tissue penetration and therefore, improved action and retention of tumor growth, by targeting miR-21 [71].

Recent studies suggest that the understanding of stereochemical effects and further modifications, such as by using phosphoryl guanidine diester (PG), can improve the pharmacological effects [72,73,74]. PG was also incorporated into the backbone of an ASO in a recent HD trial (NCT05032196) [75]. The modification itself was already introduced in an oligonucleotide in 2014 [76]. PGs have a strong nuclease resistance and a reduced toxicity, but also an increased reduction in affinity and a loss of RNase H recognition, when present in DNA gaps. With their good nuclease resistance, it might be beneficial to use this modification at the flanking regions of an ASO [41,42,43]. Used in steric block ASOs, PGs can improve the activity and uptake of the corresponding ASO and suppress target gene expression [44].

## 4. Sugar Modifications

Modifications of the sugar of a nucleotide are mostly accompanied by a modification at the 2′ C atom or a total loss of the ring structure. In both cases, the molecular character of the DNA is modified to an extent that it is not recognized by RNase H [77,78,79] (Figure 2). Therefore, such modifications are not possible in the central DNA portion of a gapmer. In general, sugar modifications in steric block ASOs, or the flanking RNA part of gapmers, improve nuclease resistance and RNA affinity [31,51]. The idea of improving the affinity of antisense strands to their target is already decades old. In 1987, Inoue et al. used a 9-nucleotide-long gapmer with two or three flanking 2′O-methylnucleotides in which the RNase H cleavage site was specifically located in the unmodified, central DNA [79]. This modification was also part of the first approved aptamer drug, pegaptanib, in 2004 [80]. Binding to RNA, this modification has minor effects on affinity, increasing melting temperature by 0 to +1.3 °C per nucleotide [27,34,38,46].

The 2′-O-methoxyethylnucleotide (2′MOE) modifications are more widely used. The 2′-MOE modifications have an even higher affinity than other modifications, as well as a good nuclear resistance, with an increase in melting temperature of +0.9 to +1.9 °C per nucleotide [34,46,52,53,81]. One of the most prominent uses of 2′MOE in ASOs is in nusinersen (Spinraza), which was FDA approved in 2016. Nusinersen is an end-to-end 2′MOE modified, 18-nucleotide-long steric block ASO that is used for the treatment of spinal muscular atrophy (SMA) [82]. A similarly prominent ASO example in which 2′MOE modifications were applied, but with a different function, is in the development of tominersen as potential treatment for Huntington’s disease (HD). The gapmer ASO, developed by Roche and IONIS Pharmaceuticals, consists of a 10-nucleotide-long DNA-gap and 5 flanking 2′-O-methoxyethylnucleotides on both wings. In March 2021, the phase III trial was stopped due to risk/benefit concerns; nonetheless, a reduction in huntingtin protein was achieved and is thus proof of the principle of using ASOs with mechanism of action via RNase H degradation [2,83,84].

In another group of sugar modifications, a linkage between the 2′-O and the 4′-C results in a C3′-endo sugar pucker of the furanose. The so-called locked nucleic acids (LNA) change the helix structure towards A-type, with respect to a DNA:RNA binding. This prevents the duplex from RNase H cleavage [85,86]. LNA:RNA bindings have a high affinity, with an increase of up to 9 °C per nucleotide with respect to DNA:RNA bindings [27,45,46,47,48,49,50]. In addition, LNA provide increased protection against endonuclease (nuclease P_1_) and 3′exonucleases (snake venom phosphodiesterase, SVPD) compared to DNA [49,51]. Three LNA modified nucleotides, each at the 3′ and 5′ ends, increase the half-life of the respective gapmer in human serum by 10-fold compared to unmodified DNA oligonucleotides [27]. LNAs also exhibit increased potency compared to 2′MOEs, although this is also associated with increased toxicity [87].

Like LNA, 2′4′-constrained 2′-O-ethyl (2′-cEt) are part of the bridged nucleic acids (BNA). The 2′-cEt have an additional methyl group at the connecting C-atom. Since the stereochemistry of the methyl group can vary in its connection to the C atom, they are referred to as enantiomers *S*-cEt and *R*-cEt [88]. Compared to LNAs, both analogs showed comparable thermal stability, potency, and increased nuclease stability against SVPD [26,51].

Modifications of the sugar are particularly good at increasing affinity and nuclease resistance. However, since they cannot be used in the DNA gap, their influence on the cleavage site and the specificity of individual bonds is limited.

## 5. Base Modifications

Base modifications may have the greatest potential to increase specificity of a single base, but they also present the greatest challenges. Although an adjustment can lead to an increase in the affinity of the base, it can also lead to an enhanced affinity for non-complementary bases, leading to an elevated likelihood of mismatches. The purpose of base modulation is to increase the specificity of the entire ASO, but most importantly, of a single base to its complementary counterpart. The smallest variants of the modification of bases are methylations, such as the 5′-methylation of cytidines (Figure 2). This methylation can reduce immunostimulatory effects, especially of CpG dinucleotides, and increase nuclease stability and thermal stability, with an increase of up to 1.1 °C per nucleotide [25,34,89]. C5′-methylations of cytidine and uracil are used throughout the molecule in some steric block ASOs, such as Spinraza, and RNase H-activating ASOs [3]. A slightly larger modification is C5′-propynyl thymine and cytosine. Compared to the single C5′-methylation, C5′propynyls increase thermal stability by +0.9 to +2.6 °C per nucleotide. Depending on their position and stereochemistry, they may even improve specificity [34,54,55,57,58,59]. The 5′thiazole pyrimidine analogues have similar characteristics for base stacking, and they improve thermal stability, with +1.7 to +2.2 °C per base [59].

A further step towards allele-selective oligonucleotide modifications could be achieved by 2′-thio pyrimidine modifications. Østergaard et al. showed that by incorporating a single 2′-thio deoxythymidine (2′-thio dT) into the DNA gap of an ASO, the affinity, but also the specificity, could be increased. For a HTT SNP-targeting ASO, the difference between melting temperatures was highest (5.6 °C) when the 2′-thio dT modification was at the position of the SNP [60]. Another well-studied pyrimidine analogue is the G-clamp, a cytosine that is scaled up to a phenoxazine with 9′-O-aminoethyl. The G-clamp and newer approaches, like G^8AE^-clamp, or guadino-G-clamp, improve binding affinity and specificity to guanine by making additional use of the Hoogsteen binding site, forming a fourth hydrogen bond to guanine [56,90,91,92]. By this, the melting temperature can be increased by +4 to +18 °C per nucleotide [46,56], especially when the G-clamp is flanked by a 5′-cytidine. In addition, the G-clamp has fluorescent properties and a 3′exonuclease resistance [56]. Increased specificity was shown in a DNA-ASO consisting of 10 nucleotides. Comparisons between 5′methylcytosine and G-clamp modifications resulted in an increase in thermal stability by 18 °C with the G-clamp. Although the melting temperature of all mismatch partners also increased compared with 5′-methylcytosine, the difference in melting temperature between match and mismatch was greatest for the G-clamp, suggesting increased specificity [55]. At the same time, the incorporation of a single G-clamp appears to have little or no effect on RNase H cleavage [46,55]. It remains to be shown if specific base modifications, like the G-clamp, can be used to improve SNP-based allele-specific targeting approaches.

## 6. Selective vs. Non-Selective Targeting Strategies

Dominant inherited diseases are often caused by heterozygous mutations, with a toxic gain-of-function. Non-selective targeting of the diseased gene would reduce both the toxic and the wild type variant and therefore, diminish the function of the protein. With a selective targeting of the mutated allele, the toxic protein load could be reduced, while preserving the physiological function of the wild type protein. Even though tominersen, a non-selective HTT ASO, showed dose-dependent reduction in HTT in the phase I/IIa trial [93], the phase III trial was stopped because of unfavorable effects outweighing potential benefits [84,94,95,96]. Previous animal studies suggested that the toxic effect of mtHTT is greater than the impairment resulting from a total loss of HTT [97]. However, it is hard to predict the ideal knockdown efficiency for nonselective degradation strategies, since it is difficult to estimate the right balance between necessary reduction in toxic mtHTT and a negative treatment response due to the loss of wtHTT [2]. Similarly, in another CAG repeat disease, SCA3, very promising non-allele-specific reduction in ATXN3 has been achieved in humanized SCA3 mice, as well as patient-derived hESCs [98,99,100]. Recently, SCA3 ASOs have just entered phase I (NCT05160558).

Table 2 gives an overview of current clinical trials using ASOs in neurodegenerative diseases caused by toxic gain-of-function mutations (reviewed in [8,101]). Treating these diseases with non-selective ASOs would result in a reduction in the mutated and wild type protein with the aim of decreasing the toxic effects. At the same time, potential protective and physiologically necessary functions of the wild type protein are also reduced. Even though some functions of proteins causing neurodegeneration are known, there are still uncertainties (Table 2). An unspecific knockdown of such a protein may have serious, unforeseeable consequences. Specifically targeting the mutant allele is supposed to result in a selective degradation of the toxic protein and preservation of the healthy allele. One of the most approachable uses of an ASO that aims for allele-specificity would be to target the mutation directly. In the case of HD or polyQ SCAs, this would be the expanded CAG repeat. An approach to circumvent the toxic effect of the expanded CAG repeat is alternative splicing. Toonen et al. masked splicing signals of the SCA3 pre-mRNA, resulting in a shortened protein version that bypasses the CAG repeat containing exon 10 and most parts of exon 11 due to an early stop codon. This modified protein showed beneficial effects on pathogenicity, with a detectable in vivo longevity of 2.5 months [102]. Even though the approach was specific for CAG toxicity, the effects were not purely allele-specific, since both alleles are targeted with these steric block ASOs. One way to directly eliminate allele-selective toxicity of the expanded CAG repeat is to use ASOs that directly target the CAG repeat at the (pre-)mRNA level. Since several proteins physiologically possess CAG repeats, a single ASO could be used for treating several different CAG repeat diseases. Likewise, the number of potential off-targets may increase, since such a CAG-specific ASO could potentially bind to several genes with CAG repeats. Evers et al. observed that such an ASO, with 21 nucleotides targeting the CAG repeat, showed allele-specific reduction not only in HTT, but also in other CAG repeat diseases, such as SCA1, 3, and DRPLA. No reduction was detected in potential CAG repeats containing off-targets, such as SCA2, ZNF384, or TATA box binding protein (TBP) [103]. Other publications with predominantly steric block ASOs show similar success in HD, SCA1, and SCA3, with dose-dependent, allele-specific reductions of up to 6.6-fold. It is believed that these steric block ASOs bind more efficient to the expanded CAG repeat containing mRNA compared to the shorter WT version, thus providing translation in an allele-specific manner [104,105,106]. A potential disadvantage of directly targeting the CAG repeat could be that every third nucleotide of the ASO that targets the CAG repeat is a guanine. Recent studies showed that in vitro and in vivo neurotoxicity is associated with the number and position of Gs within an ASO. The risk for a potential neurotoxicity increases with the number of Gs within the ASO sequence.

A less strong association to neurotoxicity was shown for the distance from the 3′-end to the first guanine. In contrast, increasing the number of As in an ASO had positive effects on the risk of potential neurotoxicity [70].

Like CAG repeat disorders, ASOs were also tested targeting other repeat diseases, such as *C9ORF72* in ALS/FTD patient cells. Different ASOs were designed targeting either the *C9ORF72* repeat (GGGGCC) itself or exonic and intronic regions up- and downstream of the repeat. Experiments on patient-derived cells and in vivo experiments showed mainly positive and allele-specific effects [130,131]. Recent publications also confirmed the possibility of achieving effective reduction via the RNase H-mediated degradation of *C9ORF72*, targeting either the repeat or intronic sequences of the gene [73,132]. Treatment of a patient with C9-ALS/FTD showed a marked reduction in polyGP-DPR (dipeptide repeat protein), with no medical or neurological adverse effects [132].

Success in patients was also reported with ASO treatment of other genes in the ALS/FTD spectrum. Treatment of an ALS patient carrying the FUS^P525L^ mutation with ION363 (jacifusen) resulted in a mutant-specific reduction of FUS protein and its aggregation, reaching clinical phase III trials (Table 2) [113]. Additionally, an ASO that targets superoxide dismutase 1 (*SOD1*) also reached phase III in clinical studies [115,130,133]. Clinical trials have also been initiated in Alzheimer’s disease using ASOs targeting *MAPT*. Preclinical studies in mice led to a significant reduction in human TAU, aggregation, and cell loss, resulting in a lifespan extension [134]. Testing of the gapmer ASO IONIS-MAPT_Rx_ (NCT03186989) in non-human primates showed a *MAPT* mRNA reduction of nearly 80%. These promising preclinical experiments resulted in the initiation of a first clinical trial for patients with mild AD [123,135]. Further potential targets for AD treatment are other genes that are directly or indirectly related to Alzheimer’s disease, such as *APP* and *APOER2*. The mRNA concentrations of these genes could also be reduced via steric block ASOs and may have beneficial effects on the course of AD [136,137].

For Parkinson’s disease, preclinical experiments in rodents achieved promising results in gapmer ASO treatment of different PD-associated genes [125,138,139]. Phase I and II clinical trials against both *LRRK2* and *SCNA* are ongoing (NCT03976349, NCT04165486).

## 7. SNP-Based Allele-Specific Treatment Strategies

Steric block ASOs that modulate splicing can be used to generate a semi-functional or non-functional protein. This shortened protein version lacks the toxic gain-of-function mutation of interest. Nevertheless, the total functional protein concentration is also lowered in this approach. Comparable effects can be achieved by non-allele-specific degradation via RNase H. Alternatively, an allele-specific degradation by targeting an SNP could surpass these non-allele-selective approaches. If the wild type allele and the disease-associated allele differ in one base at a specific site, this site might be a potential target for an allele-specific treatment strategy that aims to degrade the toxic allele at the (pre-)mRNA level. The association of a targetable SNP with the disease causing mutation might be ideal, but is not essential for such an approach. Even distant SNPs that are present in coding or non-coding regions of the disease gene far from the mutation can be used as a target to tag the disease-associated allele for degradation. The same principle can be used for steric block ASOs. Distant SNPs involved in splicing that are located in the same gene as the disease mutation could be used to disrupt the splicing machinery at the mutant allele.

In 2014, Skotte et al. published an approach to use HD-associated SNPs for allele-specific RNase H-mediated degradation of mtHTT. They evaluated 50 previously published HD-associated SNPs for targetability [54,140,141]. A total of 4 SNPs were identified in which ASOs showed increased degradation. Further optimization of the ASO sequence increased selectivity from 2.4-fold to over 100-fold and reduced the IC_50_ value against mtHTT from over 400 nM to single digit nM values for an ASO targeting the SNP rs7685686_A. To achieve this, the modifications of the flanking wings played an important role, as well as the shortening of the length of the ASO and the length of the DNA gap region. It was shown that a shorter gap, due to the shortened binding region of RNase H, resulted in enhanced allele-selective degradation. Furthermore, this study demonstrated that the SNP does not need to be in the central position of the DNA gap to achieve allele-specific degradation. By targeting the selected SNP, it would be possible to treat 48.7% of the analyzed HD population allele-specifically and another 44.9% in a non-allele-specific manner [7]. In vivo follow-up studies of this ASO in humanized HD mice (Hu97/18) confirmed the allele-specific lowering of mt*HTT* and showed positive effects in behavioral tests [142,143]. In follow-up studies, Kay et al. precisely analyzed 63 SNPs in *HTT* from people of Canadian, Swedish, French, Italian, Korean, Japanese, Chinese, and European-Canadian ethnicity, revealing three major gene-spanning haplotype groups. Differences between controls and patients of each ethnic group were used to identify the most suitable SNPs for an allele-specific treatment approach, including the analysis of the association of some SNPs to the expanded CAG repeat [144,145]. In addition to SNPs, HD-associated insertion–deletion variations (indels) can potentially be used as targets for an allele-specific approach [146]. The identification and characterization of disease-associated SNPs is a crucial step towards the design of ASOs that can potentially be used to treat the majority of patients in an allele-specific manner (Figure 3).

In a recent study, our group made use of an already known disease-associated SNP in *ATXN3* for an allele-specific ASO-based treatment in spinocerebellar ataxia type 3 (SCA3). Different ASOs were designed targeting a variant that is present in ~70% of SCA3 patients [6,147]. By establishing an in vitro platform using patient-derived iPSC-based cortical neurons, we were able to identify an ASO which leads to an allele-specific reduction in the glutamine-expanded allele by up to 75%. A one-time application showed significant allele-specific reduction for more than 7 weeks [6].

Together, these studies demonstrate that rare neurodegenerative diseases could play an important pioneering role in the development of allele-specific ASOs, especially when different modifications need to be tested in an in vitro screening. For this, an easily detectable and robust readout to quantify the allele-specific targeting efficiency is a crucial prerequisite. In the case of SCA3, iPSC-based patient-derived neuronal cultures are an ideal tool. Due to the repeat expansion, the mutant and wild type protein differ not only functionally, but also in their molecular mass. This difference can be used to display both forms separately at the protein level. Experience gained from an allele-specific design for ASOs targeting in HTT and SCA3 could then be transferred to diseases where allele-specific readout poses significant hurdles.

## 8. Target-Based ASO Design

One of the biggest challenges is to find a suitable target for an allele-specific ASO approach. If the variant that causes the disease can be directly targeted, it is usually well described. This becomes more difficult if new targets or SNPs need to be identified. As described, these can be SNPs that are located in the disease-causing gene, but are not causally related to the disease. To identify SNPs that are unrelated to the disease per se, but are in close linkage disequilibrium (LD) with a disease-associated mutation (point mutations or repeats), long-coding sequencing of larger cohorts would be of help (Figure 3). Alternatively, larger databases such as https://www.ensembl.org/index.html (accessed on 23 June 2022) or https://bravo.sph.umich.edu/freeze8/hg38/ (accessed on 23 June 2022) can be used to identify a list of candidate SNPs. Suitable new targets should have a high MAF or a high association with the disease mutation (Figure 3). A hint could be given by the MAF or the linkage disequilibrium (LD) of the potential candidate SNP with the disease-causing variant (https://ldlink.nci.nih.gov/?tab=home (accessed on 23 June 2022)) [148]. Once a list of candidate SNPs has been generated, other factors can help with further selection. The target area of the (pre-)mRNA needs to be accessible for the ASO. Computations of potential 2D structures can predict easily or poorly accessible areas (http://rna.tbi.univie.ac.at (accessed on 23 June 2022)) [149]. Short sequence patterns within an ASO are expected to have positive or negative effects on ASO activity [150]. The probability of complementary off-targets can be calculated via GGGenome (https://gggenome.dbcls.jp (accessed on 23 June 2022)). Especially for an allele-specific approach, it is advantageous if the sequence is as short as possible. With each additional nucleotide, the affinity increases, while the contribution of the individual base to the total melting temperature decreases. Thus, the influence of a single match or mismatch decreases the longer an ASO is. In addition, base variation may also play a role. Some allelic SNP constellations might be more suitable for allele-specific targeting than others. Cytosines may be more suitable as mismatch partners, as they show a greater difference in melting temperature compared to the match [151,152]. In addition, neurotoxic effects of ASOs play an important role, especially for later application. Thus, the number and position of G’s in an ASO sequence should be taken into account [70]. Once an appropriate SNP has been identified and the ASO designed, a suitable model is needed. For in vitro experiments, cells must be heterozygous for the targeted SNP and should ideally reflect the primarily affected organ/cell type. iPSC-derived cell cultures are particularly well suited for this purpose (Figure 4). They offer the advantage that they directly reflect a patient’s heterozygous genotype and can be differentiated into any cell type of interest, including neuronal subtypes affected by the disease (reviewed in [153,154]). Therapeutic effects and off-targets can easily be analyzed by transcriptomic and/or proteomic studies. (Neuro)toxicity can be quantified by, e.g., LDH assays or calcium oscillation measurements.

Another challenge is to find a suitable in vivo model, mainly mouse strains. This model should carry the target SNP on the same allele that bears the disease-associated variant. Lentiviral transduction of a humanized mutant allele with the respective target SNP has the advantage that the inserted allele and its SNPs can easily be adapted and that wild type mice can be used. Such transductions with the target allele can be performed prior to an ASO application [155,156]. In vivo mouse experiments are of particular importance for testing the in vivo functionality and acute toxicity of ASOs. Due to genomic differences between mice and humans, mouse models are less suitable for testing off-targets. Therefore, in vivo off-target studies need to be performed in non-human primates. Brain organoids from human iPSC may become increasingly important, especially in early in vitro off-target screenings (Figure 4). These self-assembled three-dimensional structures resemble the embryonic human brain and include multiple neural cell types, yielding the potential to extend the time of an ASO treatment in a multicellular system, which might be especially important for neurodegenerative diseases (reviewed in [157,158,159]).

## 9. The Challenge of Readout

In addition to the identification of appropriate SNPs, the identification of suitable readouts to prove allele-specificity is another challenge that varies in difficulty between diseases. As previously mentioned, repeat expansion disorders, such as HD [7] or DRPLA [160], bear the advantage of different molecular weight between the wild type and expanded disease protein that can be used to discriminate the respective protein. In SCA3, the wild type and mutant allele typically differ by about 40 CAG repeats, resulting in an almost 6 kDa difference in molecular weight at the protein level. Thus, both proteins can easily be separated and discriminated by Western blot testing (WB) [6,161]. In other CAG repeat expansion diseases, such as SCA1, 2, or 6, absolute size differences are smaller, making it more challenging to separate the two proteins by WB. Longer runtimes and adjustment of gel type and gradient could help in solving this problem. Alternatively, proteins or mRNA could be predigested so that the difference in glutamine repeats relative to the total peptide becomes more distinct [103]. TR-FRET assays can also be used to discriminate between mutant and wild type protein. In this assay, a fluorophore coupled antibody binds specifically to the expanded polyQ part of the mutant protein, while a second antibody binds to a different part of the protein. Only when both antibodies bind does an energy transfer occur from one antibody to the other, resulting in a fluorescent signal that can be detected and quantified. A second pair of antibodies that binds both versions of the protein is used to determine total protein concentration [162].

A more elaborated approach would be to artificially create an in vitro model by transfecting cells with either the wild type sequence or a sequence with the corresponding SNP. Subsequently, cells could be treated with ASOs, which are then examined for their allele-specific effects. A fluorescence tag could help not only to control the transfection, but also to distinguish the transfected allele from the endogenous variant. Alternatively, cells could be transfected with both variants simultaneously, if the two variants have distinct tags. This would allow for studying allele-specific effects upon an ASO treatment directly in a cell culture model and later on, in vivo. A benefit of this approach could be that multiple SNP targets could be inserted into the mutant allele variant. Therefore, cooperative ASO effects could also be part of an in vitro investigation. With this approach, multiple ASOs initiate degradation of the same target by binding to different positions.

An indirect approach to test allele specificity of an ASO could be an RNase H assay. Both variants of the target RNA need to be incubated as a duplex with the corresponding ASO (with and without mismatch) and RNase H. In a fluorometric RNase H assay, the released DNA of the gapmer binds with a N-methyl mesoporphyrin IX (NMM), forming a G-quadruplex with a fluorescence signal [163], which can be amplified by using a catalytic hairpin assembly [164]. In a next step, the efficacy could be tested in vitro and in vivo.

## 10. Conclusions

Currently, no allele-specific ASO has been approved for the treatment of a neurodegenerative disease. However, an allele-selective approach with ASOs via RNase H degradation could have great potential, especially for toxic gain-of-function mutations. If successful, the toxic allele would be eliminated, and the wild type allele could continue to maintain its physiological function. In such an approach, all coding or non-coding SNPs that are on the same allele as the disease-causing variant are potential candidates for a personalized therapeutic approach [6,144,145], (Figure 3). One reason why current research is still in its early stages might be that the allele-specific readout for most diseases has major hurdles. Repeat expansion diseases, in which a readout might be established by a different molecular weight between the wild type and mutant protein, could be flagship projects for the development of allele-specific targeting therapies using ASOs.

To standardize the identification of targetable candidate SNPs for a specific disease, long-read sequencing can help (reviewed in [165]) to directly prove the disease-association of a candidate SNP. In case of a high association of a candidate SNP with the disease-causing mutation, these SNPs would be ideal starting points to design and test allele-specific ASOs, including different modifications to increase allele-specificity. These ASOs could then be used to treat the majority of patients in an allele-specific manner.

Regarding ASO modifications, much time and effort has been spent on sugar phosphate backbone modifications. Since base modifications, such as G-clamp, can have the greatest impact on melting temperature per nucleotide [46,56] and therefore, the specificity and affinity of a single nucleotide, these modifications might be important to improve allele-selectivity. Another potential research focus for the future might be to analyze the cooperative effects of ASOs and whether there are ways to improve the administration of multiple different ASOs to enhance target engagement [166]. By administering a combination of ASOs, lower concentrations of each individual oligonucleotide could be used, reducing potential off-target effects of every single ASO. Thus, depending on the SNPs on the disease-associated allele, each patient could be given an ASO cocktail that specifically targets the mutant (pre)-mRNA for degradation through multiple binding sites.

Another hurdle is to establish a standardized workflow to identify and characterize ASOs in preclinical trials. We here propose a workflow for the design and preclinical testing of allele-specific ASOs (Figure 4). After successful target evaluation, several lead ASOs could be tested in parallel. The most effective of these will be further modified to ideally exclude toxicity and increase affinity, specificity, and longevity. Modified ASOs should then be characterized in vitro in 2D and 3D human cell culture models, as well as in vivo models. This includes effect-strength, allele-specificity, longevity, off-target effects, and (neuro)toxicity. While many challenges remain regarding allele-specific ASO therapies, allele-specific ASO strategies provide a great opportunity for future therapeutic approaches, especially for toxic gain-of-function neurodegenerative diseases.

## Figures and Tables

**Figure 1 pharmaceutics-14-01708-f001:**
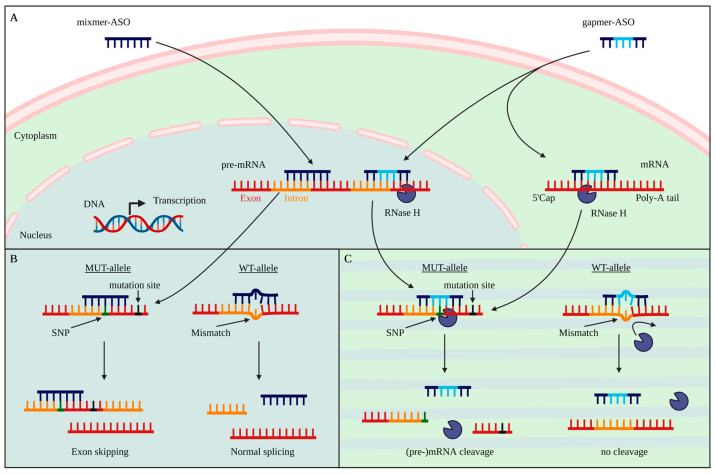
Mode of action of allele-specific antisense oligonucleotide strategies. (**A**) Mixmer ASOs and gapmer ASOs can be transported through cell membranes and nuclear pores to target (pre-)mRNA. (**B**) Mixmers act as a steric block that binds to the target pre-mRNA and can modulate the splicing process. Steric block ASOs can be used to selectively target the SNP region and therefore, modulate splicing. Binding to the WT allele has a lower affinity compared to the MUT allele, due to a mismatch. This results in a less effective splicing modulation. (**C**) Gapmer ASOs can either bind intronic regions or exonic regions of (pre)-mRNA in the nucleus or cytoplasm. Binding of gapmers with their target (pre-)mRNA recruits RNase H. The DNA:RNA binding leads to a degradation of the (pre-)mRNA. SNPs that appear on the same allele as the disease-causing mutation, such as a CAG repeat expansion, can be directly targeted with an ASO, potentially leading to a mutant-specific degradation. The mismatch on the WT allele results in a conformational change that leads to a reduced cleavage capacity.

**Figure 2 pharmaceutics-14-01708-f002:**
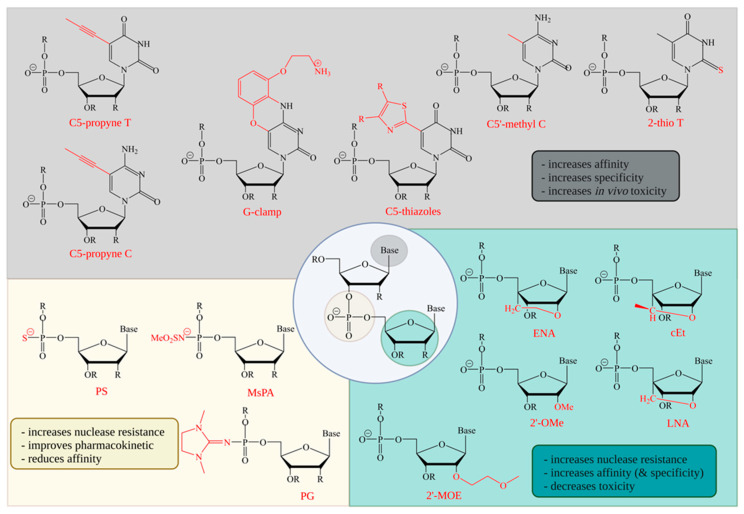
Common ASO nucleotide modifications. ASOs can be optimized at different positions of the nucleotide. Modifications at the phosphate (yellow) reduce affinity, but have positive effects on immunoreactivity and nuclease resistance and thus, also on longevity. Sugar modifications (turquoise) can increase nuclease resistance, affinity, and specificity and decrease toxicity. Disadvantageously, 2′O modifications cause the loss of deoxyribose character and cannot be recognized by RNase H. Therefore, these modifications cannot be implemented in the DNA gap region of gapmer ASOs. Base modifications (gray) may have the greatest potential for increasing the specificity of individual bases, but are also associated with an increased toxicity in vivo. cET = constrained ethylbridged nucleic acid; ENA = 2′O,4′-C-ethylene bridged nucleic acid; LNA = locked nucleic acid; MsPA = mesyl phosphoramidate; PG = phosphoryl guanidine; PS = phosphorothioate; 2′-MOE = 2′-*O*-methoxyethyl; 2′-OMe = 2′-*O*-methyl.

**Figure 3 pharmaceutics-14-01708-f003:**
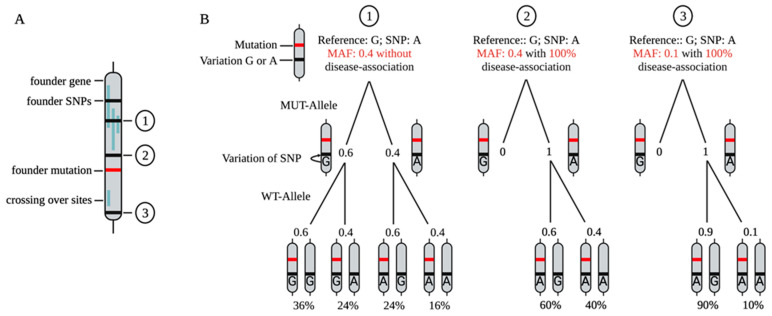
Schematic description of the association between the mutation site and non-disease-causing SNPs. (**A**) Exemplary representation of a gene with its founder mutation (red) and individual additional variants/SNPs (black). Vertical blue shadings exemplarily indicate common crossover sites. The more crossovers occur within a region, the less the association between different regions of the gene, and the linkage disequilibrium (LD) decreases. (**B**) Numbers 1, 2, and 3 show potential SNPs in the founder gene with different association strengths to the mutation site after some generations. SNP 1 has a MAF of 0.4 and is located within a region with several common crossover sites. Therefore, the SNP shows no association to the founder mutation, even though the founder gene carries SNP 1. In a patient cohort, SNP 1 is therefore similarly distributed to a control cohort. A total of 60% of the patients carry a G in SNP 1 on the same allele as the founder mutation, and 40% carry an A. The WT allele of the patients show the same distribution in SNP 1. With an ASO that specifically targets SNP 1_A, 24% of the cohort (A/G) could be treated as allele-specifical, and an additional 16% (A/A) as non-allele-specific. SNP 2 has a MAF of 0.4 but is 100% associated with the disease mutation. Therefore, every patient carries an A in SNP 2, along with the mutation on the MUT allele. With an ASO that targets SNP 2_A, 60% of the patients (A/G) could be treated as allele-specific and 40% as non-allele-specific. SNP 3 appears rarely in a population with an MAF of 0.1, but has an association of 100% to the mutation site. Every patient carries an A in SNP 3 on the same allele as the disease mutation. With an ASO that targets SNP 3_A, 90% of the patients (A/G) can be treated as allele-specific, and the remaining 10% as non-allele-specific. These example pedigrees show the importance of long coding sequencing of the whole genes of patient cohorts. Targeting rare SNPs that have a high disease association might be an optimal target. Identification of these rare, disease-associated SNPs reduces the number of ASOs needed to identify suitable ASOs for an allele-specific approach in a whole patient cohort. MAF = minor allele frequency; SNP = single nucleotide polymorphism.

**Figure 4 pharmaceutics-14-01708-f004:**
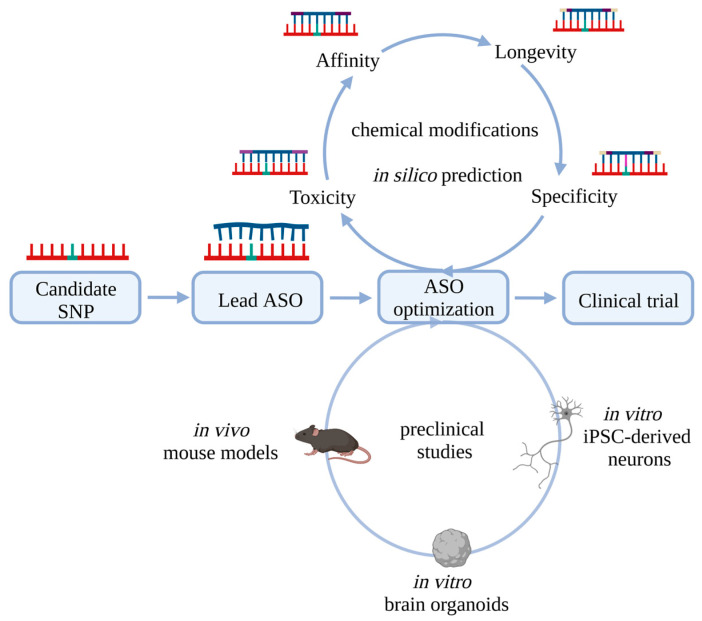
Workflow of in vitro testing and optimization of allele-specific ASOs. For allele-specific testing of ASOs, the association of the target with the mutation and the distribution in the population and patients should be known or determined. One or more lead ASOs could be tested at different positions, giving a first indication of the targetability of the (pre-)mRNA position. In further in vitro experiments, the lead ASO can be further modified to decrease toxicity and increase longevity, affinity, and allele-specificity. Different model systems have different advantages regarding evaluation of ASOs. IPSC-derived 2D models are a fast and cost-efficient tool to test the effectiveness of an ASO in a model with the genotype of a patient. Nevertheless, more complex systems, such as brain organoids, provide more precise information about the effects on the pathomechanism, such as aggregate formation, but also provide the opportunity to investigate off-targets in several neural cell types in parallel. Mice are still the model of choice to investigate (acute) toxicity and specificity in an in vivo system.

**Table 1 pharmaceutics-14-01708-t001:** Antisense oligonucleotide modifications and properties.

Modification	Tm per Nucleotide	References	Specific	General
**Backbone**				
**Phosphorothioate (PS)**	0.45 to 1 °C	[27,34,35,36,37,38,39]	Increases toxicity Improves RNase H recognition	Increases nuclease resistance Improves pharmacokinetics Reduces affinity Stereoisomers
**Mesylphosphoramidate (MsPA)**	1.3 to +1.1 °C (with respect to PS)	[39,40]	Reduces toxicity Number and position influences RNase H activation Incorporation in gap reduces protein binding Nuclease resistance (MsPA > PS)	
**Phosphoryl guanidines (PG)**	1.2 to 0 °C	[41,42,43,44]	Reduces toxicity Prevents RNase H binding, when used in DNA-gap Nuclease resistance (PG > PS) Reduced cellular uptake compared to PS	
**Sugar**				
**Locked nucleic acids (LNA)**	+1.5 to +9.1 °C	[27,45,46,47,48,49,50,51]	Can improve specificity	Increases affinity Increases nuclease resistance Decreases toxicity Not compatible with DNA-gap
**2’O-methyl (2’-OMe)**	0 to +1.3 °C	[27,34,38,46]		
**2’-O-methoxyethyl (2’-MOE)**	+0.9 to +1.9 °C	[34,46,51,52,53]	Improves cellular uptake	
**2’4’-constrained 2’-O-ethyl** **(2’-cEt)**	+4.7 to +6.1 °C	[51,54]	Can improve specificity Exonuclease resistance cET > LNA Stereoisomers Can increase specificity	
**2’-O,4’-C-ethylene-bridged** **nucleic acid (ENA)**	+5.2 °C	[49]	Exonuclease resistance ENA > LNA	
**Base**				
**G-clamp**	+4 to +18 °C	[46,55,56]	Can improve specificity Reduces RNase H activity	Increases affinity Can induce toxicity
**C5-propyne C**	+1.5 to 1.6 °C	[55,57,58,59]	Reduces RNase H activity Increases resistance	
**C5-propyne T**	+0.9 to +2.6 °C	[34,54,55,57,58,59]	Reduces RNase H activity Increases resistance	
**2-thio-thymidine**	+0.3 to +1.8 °C	[54,60]	Can improve specificity	
**5’-thiazole analogues**	+1.7 to 2.2 °C	[58,59]	RNA affinity (thiazole > propyne)	
**5-Methyl cytosine**	0 to +1.1 °C	[25,34]		

**Table 2 pharmaceutics-14-01708-t002:** Clinical trials of antisense oligonucleotides in toxic gain-of-function neurodegenerative diseases.

Disease 1 Target	Cellular Function	ASO	Phase	ASO Type/Modifications	Ref./Clinical Trial
HD–HTT	Brain development, involved in vesicle trafficking and recycling, cell division, ciliogenesis, autophagy, development [107]	Tominersen, IONIS-HTT_Rx_	Phase III halted (03/21)	Non-allele-specific, PS 2′-MOE	[2,84,108] NCT03842969
HD–HTT		WVE-003 (WVE-120101 & 120102: suspended)	Phase I/II	Allele-specific, PS stereopure	[2,72,109] NCT05032196
ALS/FTD–FUS	DNA/RNA metabolism [110]	Jacifusen/ION36	Phase III	Mutation-specific (p.P525L), PS 2′-MOE	[111,112,113] NCT04768972
ALS–SOD1	Antioxidant [114]	Tofersen/IONIS-SOD1_Rx_ (BIIB067)	Phase III	Non-allele-specific, PS 2′-MOE	[115,116] NCT02623699 NCT03070119
ALS/FTD–C9ORF72	Repeat in noncoding region [117]	IONIS-C9_Rx_ (BIIB078)	Phase I discontinued (03/22)	Non-allele specific, PS 2′-MOE	NCT03626012 NCT04288856
ALS/FTD–C9ORF72		WVE004	Phase I/II	Allele-specific (Targeting V1 and V3 transcript), PS PG stereopure	[118,119] NCT04931862
ALS/SCA2–ATXN2	RNA metabolism [120]	ION541 (BIIB105)	Phase I/II	PS 2′-MOE	[83,121] NCT04494256
AD/FTD–MAPT (TAU)	Stabilizing & promotion of microtubule assembly [122]	IONIS-MAPT_Rx_ (BIIB080)	Phase II	PS 2′-MOE	[123]
SCA3–ATXN3	Deubiquitinase [120]	ION260 (BIIB132)	Phase I	Non-allele-specific, PS 2′-MOE	[100,121] NCT05160558
PD–LRRK2	Kinase involved in lysosomal processes, autophagy, mitophagy, vesicle trafficking [124]	ION859 (BIIB094)	Phase I/II	PS 2′-MOE	[125] NCT03976349
PD–SNCA	Presynaptic protein, involved in SNARE complex assembly [126]	ION464 (BIIB101)	Phase II	PS 2′-MOE	[127] NCT04165486
Alexander disease–GFAP	Intermediate filament [128]	Zilganersen, ION373	Phase II	Non-allele-specific, PS 2′-MOE	[129] NCT04849741 CAS2305355-56-8

## Data Availability

Not applicable.

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
