# Peer review of "Towards Personalized Allele-Specific Antisense Oligonucleotide Therapies for Toxic Gain-of-Function Neurodegenerative Diseases"

_pharmaceutics, 2022, doi:10.3390/pharmaceutics14081708_

Round 1

Reviewer 1 Report

Good work!

Author Response

Thanks for reviewing.

Reviewer 2 Report

A review paper by J. Helm, L. Schöls, and S. Hauser describes the application of antisense oligonucleotides to the treatment of neurodegenerative diseases that are characterized by toxic gain of function. The work is undoubtedly interesting and may be worth publishing. Yet, despite the topic of the review is very relevant, I think it cannot be accepted in its present form and requires some definite re-writing due to the reasons summarized below.

Page 2, 1. Introduction, bottom paragraph.

“Due to their chemical properties, ASOs can diffuse freely through cell membranes and nuclear pores…”

This is a remarkable overstatement. Most of synthetic oligonucleotide analogues (including most of those pictured in Figure 2 of this paper) cannot diffuse freely through cell membranes. Analogously, not all ASO types can specifically go into nucleus. The vast field of antisense oligonucleotide and small interfering RNA cellular uptake and compartmentalization has been reviewed extensively over the last decade (see, e.g. Juliano et al., Acc. Chem. Res. 2012, 45, 1067; Juliano, et al., J. Drug Target. 2013, 21, 27; Juliano et al., Nucleic Acid Ther. 2014, 24, 101; Crooke et al. Nat. Biotechnol. 2017, 35, 230, etc.). I recommend to expand and modify the Introduction section to include more material on oligonucleotide delivery and intracellular distribution.

Page 3, 2. Antisense oligonucleotides.

“Antisense oligonucleotides (ASO) compose of a strand of DNA and RNA as initial structure that bind via Watson-Crick hybridization to its antisense RNA strand.”

Better to say “complementary RNA strand” to avoid unnecessary repetition, especially as antisense RNA is a specific term.

“Modified RNA strands and mixmers act as a steric block that modifies splicing of pre-mRNA.”

This is an oversimplification. Most of ASO types act as steric blockers, e.g. LNAs, morpholinos (PMOs), PNAs, or phosphoryl guanidines from the newer lot. Splice switching (or splice modulation) is only one manifestation, albeit an important one, of a more general mechanism. However, there are plenty of examples, including in neurology, of the use of steric blockers such as PMOs to downregulate gene expression outside of splice modulation context (see, e.g. Lu-Nguyen et al., Hum. Mol. Genet. 2021, 30, 1398-1412, doi: 10.1093/hmg/ddab136).

I suggest to include a brief discussion of a general antisense mechanism before going over to describing splice modulation in detail.

Page 4, 3. Backbone modifications

“… the sulfur of PS forms disulfide bridges with albumin …”

A remarkably bold statement, especially so as the references cited herein ([32-34]) provide no evidence whatsoever in support of such a claim. It is well and true that the current consensus says it is the propensity of phosphorothioate group (PS) in oligonucleotides to bind to proteins, which is responsible for their favorable pharmacological properties as well as for their toxicity. However, the usual understanding is that the nature of the above interaction is non-covalent (‘hydrophobic’) rather than covalent such as (transient) disulfide formation: see, e.g. ref [33], which provides an excellent overview of the subject but makes no mention of the possibility of disulfide bridges between PS ASOs and cysteine residues in proteins.

Yet, the possibility of a transient (and dynamic) disulfide formation between a PS ASO and a protein has been arguably put forth as the mechanism of their cell uptake in a recent paper by Laurent et al. (https://onlinelibrary.wiley.com/doi/10.1002/anie.202107327). I believe it is the first ever study, which provides some evidence that such a mechanism could indeed exist. Incidentally, the paper was NOT cited in the manuscript under review.

Thus, the problem of protein-ASO interaction has to be given more careful treatment, and this section ought to be re-written accordingly.

“A relatively novel modification of the backbone is mesylphosphoramidate (MsPA) …”

A general comment: If one reads the papers [41] and [42], it is easy to see that mesyl phosphoramidate modification (abbreviated as Greek letter “mu” by the authors of [41] and [42], and as MsPA by Ionis team [43]) has been designed by a single research group. It is right and fitting to give recognition to the designers of the mesyl phosphoramidate chemistry, just like it was done by Anderson et al. in their paper (ref [43], page 9028, 2nd paragraph, https://www.ncbi.nlm.nih.gov/pmc/articles/PMC8450106/pdf/gkab718.pdf).

“Slight improvement in RNase H activity could be observed when a maximum of 5 PS bonds in the DNA-gap were replaced by MsPA, especially at the 3'-gap-end. In general, MsPA seems to impede activity of the catalytic domain of RNase H rather than to promote it.”

It is worthy of note that the main impact of mesyl modification in PS ASOs is in the extended duration of the effect and reduced toxicity rather than in the intricate aspects of RNase H recruitment [43]. The same has been evidenced in [42]. The pattern of RNase H cleavage appears to be different in the case of fully PS and mixed PS/mesyl gapmers (personal observation of this reviewer), but it requires careful assessment. At the moment, the ability of mesyl ASO to support RNase H action appears to be well documented from several laboratories (see [42], [43], cf. also Hammond et al., Nucleic Acid Ther. 2021; 31:190-200, doi: 10.1089/nat.2020.0860 on the difference in 2’-deoxy and 2’-OMe/2’-MOE ASO in the nusinersen context, Fig. 2, A vs B and C; dNus-s and dNus-m both are RNase H-recruiting).

Reduced in vivo activity of fully modified mesyl ASOs vs some PS/mesyl chimaeras reported in [43] could be plausibly explained by the reduced protein binding of the former. It should be noted that in [42] cell culture experiments were carried out under lipofection conditions, which enabled the fully modified mesyl ASOs to demonstrate their advantages of longer duration and lower off-target effects. Furthermore, in Patutina et al., Proc. Natl. Acad. Sci. U.S.A. 2020, 117, 32370-32379, doi: 10.1073/pnas.2016158117 a fully modified mesyl ASO were shown to be more active than the corresponding PS ASO, when delivered by in vivo compatible cationic liposomes.

I think this paper is worth citing as well.

Perhaps, to unlock their full potential, mesyl ASOs, similarly to siRNAs, will need a suitable delivery system.

The end of the section

A general comment. Phosphoryl guanidine oligonucleotides (PGOs), which Wave team prefers to call the ‘PN’ chemistry, came from the same people as mesyls: see refs [50 and 51] (refs [45] and [47] do not relate to PGOs; it is not clear, why these were included). In fact, the very first description of phosphoryl guanidines is Kupryushkin M.S., Pyshnyi D.V., Stetsenko D.A., Acta Naturae. 2014, 6, 116-118, https://pubmed.ncbi.nlm.nih.gov/25558402/. The first evidence of biological activity (antiviral) is provided in Levina et al., Mol. Biol. 2017, 51, 633-638, doi: 10.1134/S0026893317040136, PMID: 28900092. The work by Skvortsova et al. established the mechanism of PGO action as steric block antisense, not RNase H dependent (Skvortsova et al., Front. Pharmacol., 2019, 10, 1049, doi: 10.3389/fphar.2019.01049).

These three papers need to be cited.

Two of the above works actually dealt with 2’-OMe RNA-based fully modified and charge-neutral PGOs, whereas the ref [50] describes PO or PS gapmers, where phosphoryl guanidine groups were in the 2’-OMe RNA wings. Excellent nuclease stability of PGOs was also documented therein [50].

It would not be a gross overestimation to say that, in a chemical sense, mesyl phosphoramidate and phosphoryl guanidine phosphate mimics were the most important recent addition to antisense oligonucleotide arsenal in the 2010s, which was immediately picked up by several major players in the field like Ionis Pharmaceuticals, with some candidates from Wave Life Sciences (the ‘PN’ chemistry) entering clinical trials last year.

Further comments

Table 1

General comment. It is not clear what modification the bullet points in the two columns from the right (Advantage/Disadvantage) refer to. Perhaps, it could be due to formatting of the table, but it needs to be corrected in any case. I suggest to continue horizontal lines below each modification till the right margin of the table and keep the respective advantages and disadvantages within the line.

Specific comments. Mesyl phosphoramidate (MsPA, not MePS): although ΔTm may be similar to PS, the ref [42] reports a gel shift assay (Fig. 1B), which shows that a mesyl ASO forms its duplex with RNA much faster than its PS counterpart.

Phosphoryl guanidine (PGO). The value of ΔTm of –1.2oC was taken from ref [49], which provides such data on 2’-deoxy PGOs only. The data are not particularly relevant as PG groups may not be suitable for incorporation into the gap as this may disrupt RNase H recognition. The ΔTms for the respective 2’-OMe PGOs are close to zero (personal communication of Dr. Stetsenko, the inventor of both phosphoryl guanidines and mesyl phosphoramidates). Thus, 2’-OMe PGOs may actually have better RNA affinity than isosequential PMOs.

Figure 2

It looks a bit naïve these days to import the structures straight from ChemDraw DNA template retaining all the hydrogens. I strongly recommend to strip the sugar rings from hydrogens together with the respective bonds, leaving clear Haworth formulae for ribose.

Phosphate modifications (bottom left corner): phosphoryl guanidine structure is lacking. It is given, e.g. in Levina et al. (Levina et al., Mol. Biol. 2017, 51, 633-638, doi: 10.1134/S0026893317040136, PMID: 28900092) and Kandasamy et al., Nucleic Acids Res., 2022, 50, 5443-5466, doi: 10.1093/nar/gkac018 (labeled as ‘PN’). If the authors are still unsure, the structure is provided in the attached pdf file (R = H, OMe, F).

Also, the bullet point ‘increases toxicity’, while perfectly applicable to PS, looks somewhat undeserved for mesyls as these actually decrease toxicity [42, 43] (and phosphoryl guanidines as well, by the way).

Minor points

2’-thio T is actually 2-thio T. The numeral refers to the nucleobase, not the sugar. Additionally, the structure lacks the double bond at O4.

2’-OMe. The MeO group is attached to the 2’-carbon of ribose by the wrong end.

Additional points

The paper will gain a lot from careful English language and style editing as it is in some places difficult to read (and understand, what the authors actually meant). In addition, it has plenty of typos, which need to be removed, e.g. the remains of previous reference formatting such as the message “Error! Reference source not found” throughout the text, etc.

I will be happy to re-review the manuscript after appropriate revision.

Round 2

Reviewer 2 Report

In general, I am happy with the revision of the text the authors have carried out. The manuscript now is in much better shape, and is nearly ready for publication. My further comments are relatively minor in nature, and easy to accommodate before going over to the next stage.

Yet, there remain some points that require careful attention. These are summarised below.

 Major point

Mesyl phosphoramidate discussion, starting from page 7, bottom paragraph.

Authors twice repeated the statement that the effects of mesyl phosphoramidate modification on RNase H1 mediated cleavage “are controversial”.

Please have a look at a very recent paper, which comes from the same Ionis group as Anderson, 2021:

Zhang, L.; Liang, X.H.; De Hoyos, C.L.; Migawa, M.; Nichols, J.G.; Freestone, G.; Tian, J.; Seth, P.P.; Crooke, S.T. The Combination of Mesyl-Phosphoramidate Inter-Nucleotide Linkages and 2'-O-Methyl in Selected Positions in the Antisense Oligonucleotide Enhances the Performance of RNaseH1 Active PS-ASOs. Nucleic Acid Ther. 2022 Jul 20. doi: 10.1089/nat.2022.0005. Epub ahead of print. PMID: 35861704.

It does unequivocally state that “the MsPA modification improves the RNase H1 cleavage rate of PS ASOs…”. See specifically page 7, section: “MsPA linkage modifications increase RNase H1 cleavage rate” and the Discussion section: “Our results showed the two MsPA modifications either in the gap or in the 3′ wing enhanced activities of the gap2 OMe-modified ASOs likely due to increased RNase H1 cleavage rate.”

In the light of the newly published data, I suggest the authors modify their discussion of MsPA oligos, citing the reference above and, specifically, avoiding such words as ‘controversial’.

 Minor points

1. Introduction.

The definition of ASO, I think, misses one important point, namely, noncoding RNAs, for example, microRNAs. It would be better to re-phrase it to read, e.g. “Antisense oligonucleotides (ASO) are short, modified single-stranded DNA, RNA or hybrid DNA-RNA sequences that bind complementary cellular RNAs such as (pre-)mRNAs or noncoding RNAs such as microRNAs thereby influencing their further processing. In the central dogma of protein synthesis, such an ASO (pre-)mRNA interaction would prevent the translation of a potentially toxic gene variant.”

2. Antisense oligonucleotides.

The first sentence of the paragraph is a simplified repeat of the start of the Introduction. Perhaps, authors may agree to modify the first line to read “DNA or RNA” to give just a hint at the variety of backbone structures.

Line 11 (the sentence ending with refs [25-28]).

The word ‘pure’ in conjunction with RNA should better be omitted to avoid possible association with single-stranded siRNAs. Just ‘modified RNA’ would be quite enough.

Figure 2, legend.

Top section: please correct to ‘thiazoles’.

Lower right corner, the caption in a box: please correct to read ‘increases affinity & specificity”.

I added a pdf file where I included the present review followed by my remarks on the authors' comments regarding the previous review (please see comment tabs in the text).
